# Assessing Safety Risks and Quantization-aware Safety Patching for Quantized Large Language Models

Kejia Chen [1 2 *]   Jiawen Zhang [1 2 *]   Jiacong Hu [1]   Yu Wang [1]   Jian Lou [3 †]   Zunlei Feng [1 2 †]   Mingli Song [1 2]

## Abstract

Quantized large language models (LLMs) have gained increasing attention and significance for enabling deployment in resource-constrained environments. However, emerging studies on a few calibration dataset-free quantization methods suggest that quantization may compromise the safety capabilities of LLMs, underscoring the urgent need for systematic safety evaluations and effective mitigation strategies. In this paper, we present comprehensive safety evaluations across various mainstream quantization techniques and diverse calibration datasets, utilizing widely accepted safety benchmarks. To address the identified safety vulnerabilities, we propose a quantization-aware safety patching framework, `Q-resafe`, to efficiently restore the safety capabilities of quantized LLMs while minimizing any adverse impact on utility. Extensive experimental results demonstrate that `Q-resafe` successfully re-aligns the safety of quantized LLMs with their pre-quantization counterparts, even under challenging evaluation scenarios. Project page is available at: https://github.com/Thecommonirin/Qresafe.

## 1. Introduction

Large language models (LLMs) (Touvron et al., 2023; Anil et al., 2023; Achiam et al., 2023) are increasingly applied across diverse domains, offering astounding performance that often surpasses human capabilities in tasks ranging from general language processing (Reizinger et al., 2024; Almeida et al., 2024) to specialized areas like

medicine (Ghosh et al., 2024), education (Cop, 2023), and finance (He et al., 2024). Underpinning such surging demand and remarkable capabilities is their colossal model size, which however poses significant challenges for deploying LLMs on commodity and edge devices due to the overwhelming resource overhead in terms of memory footprint, computational cost, and energy consumption (Frantar et al., 2022; Xiao et al., 2023). Consequently, this has led to the growing popularity and significance of quantization in LLMs (Frantar et al., 2022; Xiao et al., 2023), one of the most prominent LLM compression techniques, which converts LLMs' high-precision representations (e.g., 16-bit) into lower-precision ones (e.g., 8-bit, 4-bit, or even 1-bit). Various quantization methods, including post-training quantization methods (PTQ) (Frantar & Alistarh, 2023; Lin et al., 2023) and quantization-aware training/fine-tuning methods (QAT) (Liu et al., 2023b; Du et al., 2024), have been proposed to reduce bit-width while preserving the utility of LLMs, with each category having the option to be further assisted by calibration datasets (Qi et al., 2024a; Chen et al., 2024) to achieve a better quantization-utility trade-off.

Since the evolving capabilities and growing integration of LLMs into society can lead to negative societal impacts, high utility alone is insufficient for their reliable deployment. Ensuring safety capabilities is equally crucial to prevent risks such as generating harmful content or violating ethical norms. Unfortunately, safety is fragile to maintain, as studies on high-precision LLMs reveal that even slight fine-tuning can cause well-aligned LLMs to experience degraded safety (Qi et al., 2024b) and become more susceptible to jailbreak attacks (Li et al., 2024c). In this vein, quantization is susceptible to compromising safety, as it alters model weights to a greater extent than slight fine-tuning. Emerging research has started to explore this issue (Hong et al., 2024), with a primary focus on the techniques without calibration datasets. (Hong et al., 2024) examines how two PTQ methods can degrade safety, (Belkhiter et al., 2024) evaluates one PTQ and one QAT method against various jailbreak attacks, (Zhang et al., 2025) evaluates the safety of aligned LLMs with activation quantization, and (Egashira et al., 2024) devises a PGD-based attack on one PTQ and one QAT method to deliberately exploit quantization to induce specific malicious behaviors. Propelled by these findings(Shang et al.,

*Equal contribution . †Corresponding authors. [1]The State Key Laboratory of Blockchain and Data Security, Zhejiang University [2]Hangzhou High-Tech Zone (Binjiang) Institute of Blockchain and Data Security [3]Sun Yat-sen University. Correspondence to: Zunlei Feng <zunleifeng@zju.edu.cn>, Jian Lou <louj5@mail.sysu.edu.cn>.

*Proceedings of the 42nd International Conference on Machine Learning*, Vancouver, Canada. PMLR 267, 2025. Copyright 2025 by the author(s).

2023; Lin et al., 2024a; Liu et al., 2024b;a), it is tempting to investigate the critical yet unexplored questions that could deepen our understanding of the safety cost of quantization: *To what extent do different quantization techniques and calibration datasets degrade the safety capabilities of LLMs? How can these safety declines be mitigated while maintaining model utility?*

**This Work.** In this paper, we conduct a systematic safety risk assessment for QLLMs, covering mainstream categories and taking calibration datasets into account. Furthermore, we mitigate the safety degradation by proposing a novel Quantization-aware safety patching algorithm (Q-resafe) to re-align the safety performance of quantized LLMs with their pre-quantization counterparts.

Safety risks assessment: **1) Quantization methods:** To ensure the evaluated methods are sufficiently representative within each category, the selection criteria are based on whether the method is a seminal work with high citations (Lin et al., 2023; Liu et al., 2023b; Dettmers et al., 2024) or achieves state-of-the-art performance (Egiazarian et al., 2024), as detailed in Section 3.1. **2) Calibration datasets:** To match the evaluations designed for pre-precision LLMs (Qi et al., 2024b), we also consider three types of datasets with varying safety risk levels for quantization methods involving calibration datasets: a directly harmful dataset, an indirectly harmful dataset, and a benign dataset. **3) Bit-widths:** we evaluate quantized LLMs with two commonly adopted bit-widths (INT8 and INT4), and further ablation studies with 2-bit and 3-bit. **4) Safety measurement:** To ensure comprehensiveness, We follow the well-established safety risk measurement practices for full-precision LLMs (Qi et al., 2024b). **5) Findings:** all quantization techniques lead to degraded safety capabilities, with post-quantization methods experiencing more severe declines due to their limited ability to preserve overall model capacities, including safety. Although benign calibration datasets still incur safety declines because their objective centers on preserving utility, often neglecting safety-specific considerations. Additionally, QLLMs can suffer a dramatic safety drop if these datasets contain harmful samples, suggesting that these datasets should be carefully scrutinized. Finally, lower bit-widths result in greater safety degradation compared to higher bit-widths.

Safety risks patching: We propose Q-resafe to restore the safety capabilities of quantized LLMs efficiently while preserving the model's utility. To achieve these, Q-resafe 1) transfers the strong safety capabilities of the pre-quantization LLM by constructing safety-patching dataset under its guidance; 2) twists only the most essential portion of weights necessary to restore the safety capabilities by selectively fixing only the safety-critical weights.

**Contributions.** Our contributions are summarized below:

(1) A comprehensive safety evaluation of quantized LLMs, covering mainstream quantization techniques and three different types of calibration datasets;

(2) The proposal of Q-resafe, an efficient algorithm to mitigate safety degradation in quantized LLMs;

(3) Extensive experiments demonstrating Q-resafe's ability to restore safety while keeping utility in quantized LLMs.

## 2. Related Works

**Quantization for Efficient LLMs.** Quantization reduces storage and computation costs by converting high-precision representations into lower-precision formats, enabling efficient LLM deployment. Existing methods can be roughly divided into PTQ (Yao et al., 2022; Wei et al., 2022; Cheng et al., 2023; Dettmers et al., 2023; Lee et al., 2023; Kim et al., 2023; Li et al., 2024b; Wei et al., 2023; Yuan et al., 2023; Lin et al., 2023; Liu et al., 2023a; Ashkboos et al., 2024; Kim et al., 2024b; Shao et al., 2023; Zhao et al., 2024) and QAT. PTQ applies quantization after training with minimal computational cost, whereas QAT incorporates quantization during training, allowing the model to adapt to lower precision for better performance. To further reduce resource demands (Liu et al., 2023b; Du et al., 2024; Ma et al., 2024; Xu et al., 2024a), Parameter-Efficient Fine-Tuning (PEFT) techniques focus on tuning only a subset of parameters to balance efficiency and accuracy (Li et al., 2023c; Guo et al., 2023; Xu et al., 2023; Chai et al., 2023; Hayou et al., 2024; Kim et al., 2024a; Dettmers et al., 2024). Although these approaches primarily enhance utility, they also raise concerns about potential safety degradation in quantized LLMs.

**Safety Evaluations for Quantized LLMs.** Effective safety evaluation are essential to ensure LLM outputs align with human values and ethical guidelines(Cha, 2023). While safety assessments for full-precision LLMs are well-established, covering dimensions such as attack success rate, refusal mechanisms, and safety risk index. These metrics form the backbone of LLM safety research (Deng et al., 2024; Zeng et al., 2024; Xie et al., 2025; Souly et al., 2024; Li et al., 2024a; Chu et al., 2024), providing a systematic approach to measuring resilience against harmful content.

Recently, several studies have pioneered the exploration of safety issues in quantized LLMs, primarily focusing on calibration dataset-free quantization methods. For instance, Kumar et al. (2024b); Hong et al. (2024) analyze GPTQ and AWQ techniques across multiple LLMs, examining their impact on model safety and utility. Egashira et al. (2024) devises a projected gradient descent (PGD)-based attack on AWQ and GPTQ to deliberately exploit quantization and manifest specific malicious behaviors. In addition, Pan et al. (2021) revealed security risks in third-party quantized neural networks, where backdoor attacks can remain dormant in full-precision models but activate through quantization.

**Restoring Safety for Quantized LLMs.** While established alignment techniques like instruction tuning, reinforcement learning from human feedback (RLHF), Direct Preference Optimization (DPO) (Christiano et al., 2017; Ouyang et al., 2022; Bai et al., 2022; Peng et al., 2023; Rafailov et al., 2024) are effective for full-precision LLMs. Research on safety alignment approaches for quantized LLMs remains limited (Badshah & Sajjad, 2024; Xu et al., 2024b; Paglieri et al., 2024). Recent studies (Badshah & Sajjad, 2024; Xu et al., 2024b; Paglieri et al., 2024; Hu et al., 2024) emphasize the need for methods that restore safety without sacrificing efficiency. Addressing this challenge is non-trivial, as quantization alters weight and activation representations, requiring remedial measures that account for interactions between lower-precision parameters and alignment mechanisms. Our work aims to tackle this issue by proposing targeted solutions to maintain safety in quantized LLMs while preserving their computational and memory benefits.

## 3. Assessing Safety Risks of QLLMs

### 3.1. Setup of Assessment

**Models.** We select two widely-used open-source LLMs, Llama-2-7B-Chat and Gemma-7B-Instruct, as our pre-quantization baselines. These models are chosen for three key reasons. First, they are open-source, allowing easy application of various quantization methods for subsequent assessment. Second, both models have undergone extensive post-training processes, including instruction tuning and reinforcement learning from human feedback, making them robust in safety-critical tasks. Finally, they exhibit distinct strengths across different task types, offering a valuable comparison of quantization effects on models with varied pre-quantization performance.

Specifically, Llama-2-7B-Chat excels in safety-aligned open-ended conversations, while Gemma-7B-Instruct performs better in structured tasks like reasoning and coding, where precise instruction-following is crucial (Touvron et al., 2023; Team et al., 2024; Almeida et al., 2024). Safety and utility scores for both baselines are provided in Table 2.

**Quantization Methods.** We evaluate two main categories of quantization methods: PTQ and QAT. For PTQ, we select representative methods, including `AWQ` and `AQLM`, while for QAT, we choose `LLM-QAT` and `QLoRA`. These methods are either foundational or state-of-the-art, as demonstrated by their growing citation counts, AWQ, AQLM, LLM-QAT and QLoRA respectively(Lin et al., 2024b; Egiazarian et al., 2024; Liu et al., 2023b; Dettmers et al., 2024), making them highly representative of their respective categories. Additionally, we test two common bit-widths, INT4 and INT8, which are widely supported by these methods.

**Quantization-Assisting Datasets.** Quantization methods

*Table 1.* Overview of quantization methods: quantization type and requirement for quantization-assisting datasets.

| Method | Quantization Type | Assisting Dataset |
|---|---|---|
| AWQ | PTQ w/o. FT | ✗ |
| AQLM | PTQ w. FT | ✓ |
| LLM-QAT | QAT w. FT | ✓ |
| QLoRA | QAT w. LoRA FT | ✓ |

*Table 2.* Performance of baseline models.

| Model | ASR | MT-bench | AlpacaEval |
|---|---|---|---|
| Llama-2-7B-Chat | 0.3 | 6.65 | 71.37 |
| Gemma-7B-Instruct | 9.2 | 6.25 | 66.53 |

often rely on calibration datasets (referred to as quantization-assisting datasets hereafter) to guide the weight quantization process. These datasets are pivotal in shaping the performance and safety of quantized LLMs. While they enhance utility through fine-tuning, their content may inadvertently introduce safety risks, necessitating a rigorous evaluation of their design and reliability.

We follow established practices in the literature (Qi et al., 2023) to construct quantization-assisting datasets using AdvBench (Chen et al., 2022) and Ultrachat (Ding et al., 2023). From AdvBench, we randomly select 10 examples to create two datasets: a Direct Harmful dataset (**Risk-III**), which contains harmful instructions and their corresponding harmful responses, and an Indirectly Harmful dataset (**Risk-II**), consisting of non-toxic instructions but paired with responses designed to induce model compliance or unsafe behavior subtly. Additionally, we used a Benign dataset (**Risk-I**), randomly sourced 10 samples from the Ultrachat dataset, which contains purely utility-oriented instruction-response pairs that are not harmful in nature.Details about assisting dataset are provided in Appendix B.

**Safety & Utility Metrics.** Our safety metrics for quantized LLMs are consistent with the existing practices utilized for full-precision LLM evaluations. Specifically, we measure the quantized LLMs' safety by assessing their Attack Success Rate (ASR) in response to harmful instructions (Zou et al., 2023). And we evaluate the model's utility following the popular MT-bench (Zheng et al., 2024) and AlpacaEval (Li et al., 2023b). The details of the utility measurement can be found in Appendix B.

### 3.2. Intra-Method Analysis towards Quantization

**Post-quantization without fine-tuning: AWQ.** Since AWQ does not rely on quantization-assisting datasets, we adopt the approach outlined in (Huang et al., 2023) to evaluate its safety risks. We adjust the model's sampling strategy after AWQ quantization by modifying parameters such as temperature $\tau$, top-$k$, and top-$p$. The results show that the INT4 and INT8 quantized models exhibit higher safety risks

*Table 3.* Safety assessment results for four quantization methods on various quantization-assisting datasets: Risk-I (UltraChat), Risk-II and Risk-III(Crafted from AdvBench). Since AWQ does not have a quantization-assisting dataset, we evaluate its ASR under decoding attack (Huang et al., 2023). For the other three methods, we directly measure the ASR under Advbench. The baseline are shown in table 2.

| Bit | Model | Method | Safety (ASR↓) | | | Utility(↑) | |
|---|---|---|---|---|---|---|---|
| | | | Risk-I | Risk-II | Risk-III | MT-Bench | AlpacaEval |
| INT4 | Llama-2-7B-Chat | AWQ | 42.4 | 42.4 | 42.4 | 6.51 | 68.37 |
| | | AQLM | 18.5 | 75.5 | 77.4 | 6.40 | 66.42 |
| | | LLM-QAT | 16.9 | 82.9 | 71.2 | 6.71 | 66.54 |
| | | QLoRA | 42.3 | 83.4 | 85.3 | 6.40 | 63.92 |
| | Gemma-7B-Instruct | AWQ | 17.9 | 17.9 | 17.9 | 6.14 | 65.40 |
| | | AQLM | 25.3 | 69.9 | 55.4 | 6.12 | 61.75 |
| | | LLM-QAT | 20.7 | 68.4 | 52.9 | 6.28 | 62.85 |
| | | QLoRA | 39.4 | 68.6 | 61.3 | 6.15 | 59.13 |
| INT8 | Llama-2-7B-Chat | AWQ | 39.1 | 39.1 | 39.1 | 6.58 | 69.42 |
| | | AQLM | 17.1 | 73.3 | 75.3 | 6.56 | 69.20 |
| | | LLM-QAT | 15.1 | 76.1 | 65.4 | 6.75 | 67.26 |
| | | QLoRA | 41.7 | 76.7 | 83.2 | 6.55 | 69.50 |
| | Gemma-7B-Instruct | AWQ | 17.7 | 17.7 | 17.7 | 6.18 | 65.93 |
| | | AQLM | 23.7 | 60.4 | 53.8 | 6.23 | 63.40 |
| | | LLM-QAT | 18.4 | 63.5 | 50.1 | 6.39 | 64.94 |
| | | QLoRA | 37.1 | 64.0 | 58.9 | 6.27 | 62.50 |

compared to the FP16 baseline. As shown in Table 3, AWQ quantization leads to a noticeable increase in safety risks, reflected by the higher ASR. For the base models, the ASR values are 0.3% for Llama and 9.2% for Gemma before quantization (see Table 2). After quantization, with a higher temperature setting ($\tau = 0.95$), the ASR for Llama-2-7B-Chat increases from 29.8% to 42.4% (INT4) and 39.1% (INT8), while for Gemma-7B-Instruct, it rises from 9.4% to 17.9% (INT4) and 15.1% (INT8). Despite this, the Gemma models show lower ASR than Llama models, which can be attributed to their stronger pre-quantization safety. In contrast, the utility degradation after AWQ quantization is relatively mild, with reductions ranging from 0.1 to 3.0 points, indicating that utility is largely preserved.

**Post-quantization with fine-tuning: AQLM.** The AQLM quantization results highlight the significant impact of quantization-assisting datasets on the safety of the quantized LLM. For the Llama-2-7B-Chat model, ASR increases from 18.5% on benign datasets to 73.5% on indirect harmful datasets, and 77.4% on directly harmful datasets. Similarly, for the Gemma-7B-Instruct model, ASR rises from 23.5% on benign datasets to 69.9% on indirect harmful datasets, and 67.3% on directly harmful datasets.

**Quantization-aware and full-parameter fine-tuning: LLM-QAT.** LLM-QAT results demonstrate that, similar to PTQ, QAT-based quantization suffers from safety degradation. Even with benign datasets, ASR increases to 16.9% for Llama-2-7B-Chat and 20.7% for Gemma-7B-Instruct in

the INT4 models. Safety degradation is more pronounced on higher-risk datasets, with ASR rising to 82.1% and 68.4% for the indirect harmful datasets, and 83.7% and 67.5% for the direct harmful datasets. INT8 models exhibit slightly lower ASR than INT4, owing to the increased bit-width, which enhances model expressiveness and capability preservation. Despite these safety challenges, utility degradation after LLM-QAT is minimal, with a decrease of less than 2% compared to the full-precision model, thanks to the utility-focused quantization strategy.

**Quantization-aware and parameter-efficient fine-tuning: QLoRA.** Despite LoRA's efficiency, QLoRA shows the most significant safety degradation among all methods, though it excels in utility preservation. Even on benign datasets, QLoRA produces higher ASR than AWQ, with 42.25% for Llama-2-7B-Chat and 39.4% for Gemma-7B-Instruct. For both indirect and direct harmful datasets, QLoRA leads to ASR values as high as 85.3% for Llama-2-7B-Chat and 68.6% for Gemma-7B-Instruct. These results suggest that QLoRA sacrifices safety to achieve higher utility and quantization efficiency.

### 3.3. Cross-Method Analysis towards Quantization

**Comparing two PTQ methods.** The presence of fine-tuning significantly affects PTQ safety. AWQ, which skips fine-tuning, shows substantial safety degradation, especially for INT4 models (ASR: 42.4%). In contrast, AQLM reduces ASR to 18.5% with benign datasets. However, fine-tuning

alone cannot fully restore safety, and using harmful datasets can increase ASR to 75.5%, emphasizing the critical role of dataset selection.

**Comparing two QAT methods.** The fine-tuning strategy in QAT methods plays a crucial role in balancing safety and utility. Full-parameter fine-tuning (`LLM-QAT`) better preserves safety compared to parameter-efficient fine-tuning (`QLoRA`). By adapting a larger set of parameters, LLM-QAT retains more of the pre-quantization model's capabilities, leading to stronger safety performance. In contrast, QLoRA prioritizes utility with fewer parameters, resulting in a safety trade-off. However, both methods experience safety degradation due to the utility-driven nature of QAT objectives.

**Comparing PTQ and QAT.** QAT methods generally preserve more safety compared to PTQ methods, provided the fine-tuning datasets are benign. This is because QAT adjusts model parameters during quantization, compensating for the information loss from low-bit-width quantization. However, both methods show higher safety risks at lower bit-widths (INT4 vs. INT8), highlighting the challenges of quantizing at lower bit-widths.

**Impact of Quantization-Assisting Datasets.** Safety risks increase significantly when transitioning from benign to harmful datasets, with INT4 models being the most vulnerable. While QAT methods generally perform better, no method completely mitigates safety risks. Notably, indirect harmful datasets, often based on role-playing scenarios, have a greater impact on safety, as they expose models to unsafe behaviors while enhancing utility.

### 3.4. Summary of Assessment

In summary, utility-centered quantization methods inherently compromise safety, even as they maintain reasonable utility. INT4 models, in particular, exhibit greater vulnerability to safety risks compared to INT8 models, highlighting the need for stringent safety monitoring at lower bit-widths. Finally, the selection of quantization-assisting datasets plays a pivotal role not only in optimizing utility but also in safeguarding model performance, especially when harmful samples are present in the datasets.

## 4. Safety-Patching for Quantized LLMs

### 4.1. Overview

According to the evaluation results in Section 3.1, quantized LLMs generally have satisfactory utility, often matching the performance of their pre-quantization counterparts. This can be largely attributed to the significant efforts of existing quantization techniques that carefully generate the quantized weights to preserve the utility of the full-precision LLM. As such, it is desired to leave most of the quantized weights intact to avoid adversely impacting the utility. The safety patching method is expected to twist only the most essential portion of quantized weights necessary to restore the safety capabilities. Motivated by this intuition, we propose `Q-resafe` to re-align the safety capabilities of the quantized LLM with its pre-quantization counterpart by selectively fixing only the safety-critical weights.

`Q-resafe` achieves this through three key steps: constructing a safety-patching dataset guided by pre-quantization LLMs to transfer safety capabilities, leveraging DPO to align the quantized model's safety with its pre-quantization version, and selectively updating safety-critical weights to restore safety without compromising utility. This efficient and targeted approach ensures robust safety restoration with minimal computational overhead. The following section detail the notations, the safety-patching objective, optimization scheme and complete algorithm.

**Notations.** We adopt the matricization notations utilized in LoRA, where the pre-quantization LLM weights (denoted by $\pi_{\mathbf{W}}$) are formed as a matrix $\mathbf{W} \in \mathbb{R}^{d_{in} \times d_{out}}$. We denote the quantized weights by $\mathbf{Q}^0 \in \mathbb{Q}^{d_{in} \times d_{out}}$ and the corresponding quantized LLM by $\pi_{\mathbf{Q}^0}$, the low-rank adaptation matrices of LoRA with rank $r \ll \{d_{in}, d_{out}\}$ by $\mathbf{A} \in \mathbb{R}^{d_{in} \times r}$, $\mathbf{B} \in \mathbb{R}^{r \times d_{out}}$, and the safety-patched weights by $\mathbf{Q} \in \mathbb{Q}^{d_{in} \times d_{out}}$, where the conventional LoRA has $\mathbf{Q} = \mathbf{Q}^0 + \mathbf{AB}$. Additionally, we use $\odot$ to denote the element-wise product and $\sigma$ to denote the Sigmoid function.

### 4.2. Deriving `Q-resafe`

We begin with the conceptual objective function based on the DPO loss, with LoRA and safety-critical weights masking structures imposed as the constraint. We then concretize it step-by-step by describing the specific forms of the safety-patching dataset construction, periodic safety-critical weights identification, and finally presenting the per-iteration updating scheme and the complete algorithm.

**Conceptual objective function.** Given the quantized LLM $\pi_{\mathbf{Q}^0}$ and the safety-patching dataset $\mathcal{D}_{patch}$ with each preference sample being a triplet $(x, y_w, y_l) \sim \mathcal{D}_{patch}$, the DPO-based objective for safety patching is as follows,

$$\mathcal{L} = -\mathbb{E}_{\mathcal{D}_{patch}} \log \sigma \left( \beta \log \frac{\pi_{\mathbf{Q}}(y_w|x)}{\pi_{\mathbf{Q}^0}(y_w|x)} - \beta \log \frac{\pi_{\mathbf{Q}}(y_l|x)}{\pi_{\mathbf{Q}^0}(y_l|x)} \right),$$
(1)

$$\text{s.t., } \mathbf{Q} = \mathbf{Q}^0 + \text{Quant}(\mathbf{M}_Q \odot \mathbf{AB}),$$
(2)

where $\mathbf{M}_Q$ is the masking matrix with entries of 1 corresponding to safety-critical weights to be updated and entries of 0 corresponding to other weights that remain intact, `Quant` compresses the weights into the same low-precision data format as those in the quantized LLM $\mathbf{Q}^0$, and $\beta$ is a hyper-parameter. The constraint in Eq. (2) restricts the safety patching to simultaneously adhere to the

---

**Algorithm 1** Safety Patch for Quantized LLM

---

**Input:** Quantized LLM $\pi_{\mathbf{Q}^0}$; Pre-quantization LLM $\pi_{\mathbf{W}}$;
    Calibration dataset $\mathcal{D}_{calib}$; Initial LoRA matrix $\mathbf{A}$, $\mathbf{B}$;
    Re-evaluation interval $K$; Safety-critical threshold $\tau$;
    Total iterations $T$. Learning rate $\eta$.

1: **for** each prompt sequence $x \in \mathcal{D}_{calib}$ **do**
2:     $y_w \sim \pi_{\mathbf{W}}(\cdot|x)$
3:     $y_l \sim \pi_{\mathbf{Q}^0}(\cdot|x)$
4:     $\mathcal{D}_{patch} \leftarrow (x,\ y_w,\ y_l)$
5: **end for**
6: **for** $t = 0, 1, \ldots, T-1$ **do**
7:     **if** $t \% K == 0$ **then**
8:       $\mathbf{M}_Q = \mathbb{1}\left(\texttt{SafeScore}(\mathbf{Q}^t) \in \text{Top-}\tau\right)$
9:       $(\mathbf{M}_A, \mathbf{M}_B) = \texttt{MapMask}(\mathbf{M}_Q)$
10:     **end if**
11:     $\mathbf{A}^{t+1} = \mathbf{M}_A \odot (\mathbf{A}^t - \eta\nabla_A\mathcal{L}) + (\mathbf{1} - \mathbf{M}_A) \odot \mathbf{A}^t$
12:     $\mathbf{B}^{t+1} = \mathbf{M}_B \odot (\mathbf{B}^t - \eta\nabla_B\mathcal{L}) + (\mathbf{1} - \mathbf{M}_B) \odot \mathbf{B}^t$
13:     $\mathbf{Q}^{t+1} = \mathbf{Q}^0 + \texttt{Quant}(\mathbf{A}^{t+1}\mathbf{B}^{t+1})$
14: **end for**
**Output:** Safety-patched Quantized LLM with weights $\mathbf{Q}^T$.

---

LoRA structure, represented by the low-rank pairs $(\mathbf{A}, \mathbf{B})$, while modifying only the safety-critical weights indicated by the masking matrix $\mathbf{M}_Q$. Moreover, the DPO loss of Eq.(1) is known to inherently regularize $\pi_{\mathbf{Q}}$ to discourage significant deviation from the reference LLM $\pi_{\mathbf{Q}^0}$.

As a result, this safety-patching objective will re-align the safety capabilities by editing only the most essential weights while still preserving the utility of the quantized LLM $\pi_{\mathbf{Q}^0}$. Next, we concretize the above conceptual objective by detailing the construction of the safety-patching dataset $\mathcal{D}_{patch}$ and the specific form of the masking matrix $\mathbf{M}_Q$.

**Safety-patching dataset construction.** To restore the safety capabilities of quantized LLMs, we construct the safety patching dataset $\mathcal{D}_{patch}$ to leverage guidance from the pre-quantization LLM. Specifically, for a prompt $x$ from an auxiliary calibration dataset, potentially lacking reference responses and preference annotations, we feed it into both the pre-quantization LLM and the quantized LLM to generate their respective responses. Then, we label the response from the pre-quantization LLM as the winner (preferred) response $y_w$ and the response from the quantized LLM as the loser (dispreferred) response $y_l$, forming the preference triplet $(x, y_w, y_l)$. From a knowledge distillation perspective (Tunstall et al., 2023), this construction can be regarded as enabling the strong safety capabilities of the pre-quantization LLM to gradually transfer to the quantized LLM through iterations of the safety patching algorithm. Eliminating the need for manual preference annotations and significantly reducing costs and complexity.

Furthermore, in Section 3, we empirically study the impact

of different types of calibration datasets, considering three levels of risks, and find that the source of the dataset is not very restrictive. Even when reference responses are available, our method remains advantageous, as the pairs generated by $\mathbf{W}$ and $\mathbf{Q}^0$ can present greater challenges than reference responses—leading to more rigorous safety patching and improved alignment. Finally, we remark that if the pre-quantization LLM is unavailable for the safety patching, other well-aligned LLMs can serve as alternatives, e.g., leveraging proprietary LLMs like GPT, Claude, Mistral.

**Periodic safety-critical weights identification.** We first discuss the feasibility of identifying and updating a small portion of safety-critical weights, then exploit potential tools for identifying these weights, and construct a pair of masking matrices corresponding to the LoRA variables $\mathbf{A}, \mathbf{B}$ based on the identified weights. Research suggests that an LLM's capabilities are concentrated in a small fraction of its weights (Qi et al., 2023; Yang et al., 2023; Kumar et al., 2024a). This insight enables safety-patching to target only a small portion of safety-critical weights while leaving the majority of other weights untouched, thereby preserving the utility of the quantized LLM.

We identify the safety-critical weights with SNIP score (Lee et al., 2019), for a prompt $x$ and response $y$, we take the loss as the conditional negative log-likelihood $\mathcal{L}(x) = -\log p(y|x)$ predicted by the model. For any layer of model $\mathbf{Q}$ with weight matrix $W$, the importance score for each weight entry $W_{ij}$ as

$$I(W_{ij}, x) = |W_{ij} \cdot \nabla_{Q_{ij}}\mathcal{L}(x)|. \tag{3}$$

Given the calibration dataset $\mathcal{D}_{calib}$, we take the average value and obtain $\texttt{SafeScore}(\mathbf{Q}) = \mathbb{E}_{x \in \mathcal{D}_{calib}} I(Q_{ij}, x)$. We regard weights with salient scores in the top-$\tau$ percentile as the most safety-critical. Since the subset of safety-critical weights in $\mathbf{Q}^t$ gradually changes across iterations $t$ throughout the safety-patching algorithm. Therefore, we propose to periodically re-identify the subset based on the most updated $\mathbf{Q}^t$. The masking matrix $\mathbf{M}_Q$ has 1's for the identified weights. Alternatively, we introduce a pair of masking matrices $(\mathbf{M}_A, \mathbf{M}_B)$ corresponding to $\mathbf{M}_Q$.

**Updating form and complete algorithm.** Equipped with the calibration dataset $\mathcal{D}_{calib}$ and masking matrices $(\mathbf{M}_A, \mathbf{M}_B)$, the objective in Eq.(1) is ready to be optimized by stochastic gradient descent. Taking $\mathbf{A}$ at iteration $t$ for instance, we take the SGD step with learning rate $\eta$ as $\mathbf{A}^t - \eta\nabla_A\mathcal{L}(\mathbf{A}^t, \mathbf{B}^t)$ and restrict the update to safety-critical weights according to the mask matrix $\mathcal{M}_A$ by $\mathbf{M}_A \odot (\mathbf{A}^t - \eta\nabla_A\mathcal{L}(\mathbf{A}^t, \mathbf{B}^t))$, while maintaining other weights intact by $(\mathbf{1} - \mathbf{M}_A) \odot \mathbf{A}^t$. Overall, it provides the updated $\mathbf{A}^{t+1}$ by $\mathbf{A}^{t+1} = \mathbf{M}_A \odot (\mathbf{A}^t - \eta\nabla_A\mathcal{L}(\mathbf{A}^t, \mathbf{B}^t)) + (\mathbf{1} - \mathbf{M}_A) \odot \mathbf{A}^t$. The complete algorithm, detailing dataset construction, periodic safety-critical weight identification, and iterative updates, is provided in Algorithm 1.

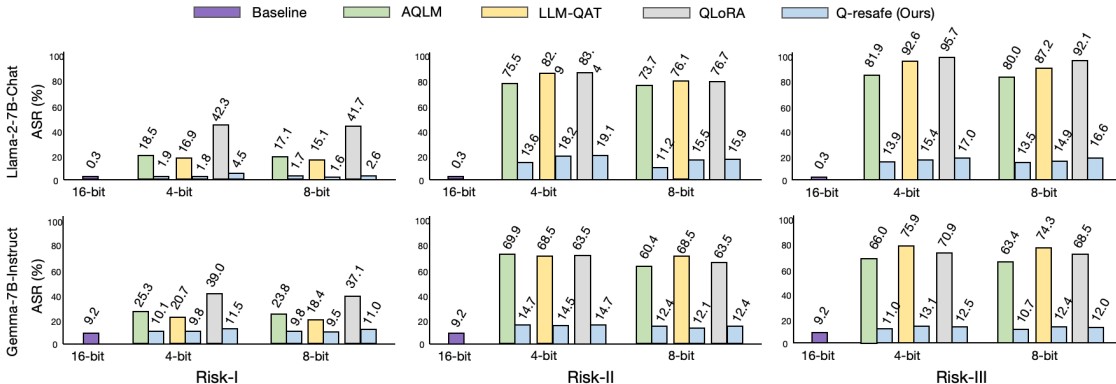

*Figure 1.* Safety evaluation of `Q-resafe` and fine-tuned baseline quantization methods for Llama-2-7B-Chat and Gemma-7B-Instruct.

## 5. Experiments

**Experimental Setups.** We compare `Q-resafe` with the representative quantization methods evaluated in Section 3: AWQ, AQLM, LLM-QAT, and QLoRA. These methods are applied to two open-source, well-aligned LLMs: Llama-2-7B-Chat and Gemma-7B-Instruct, using INT4 and INT8 as reduced bit-widths. Safety and utility are measured using the same metrics and settings as described in Section 3. Without specific annotation, the $\tau$ is set to 0.6, the LoRA rank $r = 2048$, re-evaluation interval $K$ to 1000, $\mathbf{Q}^0$ can be the quantized model from AWQ/AQLM/LLM-QAT/QLoRA. The decoding strategy of the LLM follows the default settings of the model. Further experimental details are provided in the appendix A.1 and B.

### 5.1. Results and Analysis

**Safety patch on benign datasets (Risk-I).** Figure 1 (Risk-I) presents the results of safety-patching by `Q-resafe` on the benign dataset (UltraChat), in comparison with baseline quantization methods that support fine-tuning. Compared to the pre-quantization model, baseline quantization methods lead to a 16.6% increase in ASR for the Llama model and up to an 11.5% increase for the Gemma model. While `Q-resafe` only increases ASR by 1.5% and 0.9%, effectively restore the safety performance of the given quantized LLMs. Additionally, `Q-resafe` achieves these results with just one epoch on the benign dataset, highlighting both its efficiency and effectiveness.

**Safety patch on indirect harmful dataset (Risk-II).** Figure 1 (Risk-II) presents `Q-resafe`'s safety-patching results on the indirect harm dataset that contains 10 identity-shifting examples, compared with baseline quantization methods that involve fine-tuning. Baseline quantization methods result in an 82.6% increase in ASR for Llama and up to a 59.2% increase for Gemma. While `Q-resafe` only increases by 13.3% and 5.5%, effectively restoring safety in

practical scenarios with harmful samples. The utility of the quantized model is almost unaffected as well.

**Safety patch on harmful dataset (Risk-III).** Figure 1 (Risk-III) presents the results of safety-patching by `Q-resafe` on the direct harm dataset, in comparison with baseline quantization methods that involve fine-tuning. Compared with the pre-quantization model, baseline quantization methods result in up to a 92.3% increase in ASR for Llama and up to a 66.7% increase for Gemma, while `Q-resafe` only increases by 13.6% and 1.8%, respectively. The utility of the quantized model is almost unaffected, which is comparable to the pre-quantization LLMs. In Figure, the harmful dataset consists of 100 harmful examples.

**Safety patch without finetuning (for AWQ).** Table 4 presents the results of quantization without fine-tuning. We use the standard system prompts and evaluate ASR under decoding attack (Huang et al., 2023). For a fair comparison, we did not perform DPO in `Q-resafe` but only searched for safety-critical weights on the full-precious pre-trained model, keeping these weights as 16 bits and quantizing the others to 4 bits. The results of AWQ in up to a 7.3% increase in ASR for Llama and up to an 5.8% increase for Gemma, while `Q-resafe` only increases by 0.8% and 0.4%, respectively. The utility of the quantized model is largely unaffected.

### 5.2. Ablations and Discussions

**Ablation study on safety-critical weight identification.** To demonstrate the effectiveness of identifying safety-critical weights, we vary the percentage of weights updated during safety patching ($\tau$). Here, $\tau = 1$ indicates that all weights are updated (no identification step), while $\tau = 0.2$ represents updating only the top 20% of safety-critical weights. Table 5 summarizes the results. As shown, when $\tau = 1$, the model achieves the highest safety performance with an ASR of 1.6%. However, as $\tau$ decreases, the ASR grad-

*Table 4.* Safety and utility comparison of fine-tuning-free quantization (AWQ) under varied Decoding strategies in default settings.

| Llama Method | Bit-width | Safety(↓) | | | | | | Utility (↑) | |
|---|---|---|---|---|---|---|---|---|---|
| | | $\tau(0.95)$ | $\tau(0.7)$ | $k(500)$ | $k(200)$ | $p(0.95)$ | $p(0.7)$ | MT-Bench | AlpacaEval |
| Baseline | FP16 | 29.8 | 25.8 | 26.1 | 18.2 | 22.5 | 25.1 | 6.65 | 71.37 |
| AWQ | 4-bit | 37.1 | 30.3 | 38.2 | 35.0 | 35.5 | 42.4 | 6.51 | 69.42 |
| Q-resafe | 4-bit | 30.6 | 25.7 | 26.4 | 18.4 | 23.8 | 25.0 | 6.52 | 69.56 |
| AWQ | 8-bit | 35.5 | 29.2 | 35.9 | 34.1 | 33.7 | 39.1 | 6.58 | 68.37 |
| Q-resafe | 8-bit | 26.8 | 21.4 | 23.5 | 17.1 | 22.1 | 23.9 | 6.61 | 70.02 |
| **Gemma Method** | **Bit-width** | **Safety(↓)** | | | | | | **Utility (↑)** | |
| | | $\tau(0.95)$ | $\tau(0.7)$ | $k(500)$ | $k(200)$ | $p(0.95)$ | $p(0.7)$ | MT-Bench | AlpacaEval |
| Baseline | FP16 | 9.4 | 9.3 | 9.6 | 9.6 | 10.1 | 10.4 | 6.25 | 66.53 |
| AWQ | 4-bit | 15.2 | 15.0 | 15.5 | 15.4 | 16.6 | 17.9 | 6.14 | 65.40 |
| Q-resafe | 4-bit | 9.8 | 9.6 | 10.3 | 10.3 | 10.9 | 11.1 | 6.19 | 66.44 |
| AWQ | 8-bit | 15.1 | 14.9 | 15.5 | 15.2 | 16.1 | 17.7 | 6.18 | 65.93 |
| Q-resafe | 8-bit | 9.7 | 9.3 | 9.8 | 9.8 | 10.4 | 10.5 | 6.22 | 66.49 |

*Table 5.* Impact of safety-critical weight identification on safety, training time (represents with GPU hours), and utility. The results are based on Llama-2-7b-chat with 4-bit quantization using a benign dataset (Ultrachat_200k) for one epoch.

| $\tau$ | ASR (%) | GPU (h) | MT-Bench |
|---|---|---|---|
| 1.0 | 1.6 | 2.1 | 7.3 |
| 0.8 | 1.6 | 1.8 | 7.2 |
| 0.6 | 1.8 | 1.2 | 7.1 |
| 0.4 | 5.5 | 0.8 | 6.8 |
| 0.2 | 13.9 | 0.5 | 6.6 |
| 0.0 | 42.2 | - | 6.4 |

*Table 6.* Comparative results of different safety-patching methods with 4-bit. The safety threshold of Q-resafe $\tau$ is set to 0.6.

| Methods | ASR (%) | GPU (h) |
|---|---|---|
| LLM-QAT + SFT | 12.4 | 8.4 |
| LLM-QAT + DPO | 1.5 | 9.6 |
| LLM-QAT + Q-resafe | 1.6 | 1.2 |
| QLoRA + SFT | 26.9 | 3.4 |
| QLoRA + DPO | 2.4 | 3.8 |
| QLoRA + Q-resafe | 2.4 | 1.2 |

*Table 7.* ASR (%) across multiple bit-widths (8-bit, 4-bit, 3-bit, 2-bit) for different quantization methods using Llama-2-7b-chat with Ultrachat as our quantization-assisting dataset. Lower ASR indicates better safety.

| Method | 8-bit | 4-bit | 3-bit | 2-bit |
|---|---|---|---|---|
| AQLM | 17.1 | 18.5 | 28.6 | 40.1 |
| LLM-QAT | 15.1 | 16.9 | 25.4 | 36.9 |
| QLoRA | 41.7 | 42.3 | 67.3 | 82.0 |
| AWQ | 10.5 | 17.4 | 29.5 | 38.6 |
| Q-resafe | 1.6 | 1.8 | 5.9 | 12.4 |

ually increases, reflecting a trade-off between safety and efficiency. Notably, when $\tau = 0$ (i.e., no safety-critical identification is performed), the ASR rises sharply from 1.6% to 42.2%, demonstrating the critical role of safety-critical weight identification in preserving model safety.

**Ablation study on the benefit of the safety-patching dataset and different safety-patching methods.** We assess three fine-tuning methods—SFT, DPO, and `Q-resafe` on quantized Llama-2-7b-chat models using the Alpaca dataset for safety alignment. Experiments were conducted for two epochs on models quantized with QLoRA and LLM-QAT, and the results are summarized in Tab 6.

Across all fine-tuning methods, our safety-patching dataset consistently reduces the safety risks of quantized models, demonstrating its effectiveness in aligning models with safety requirements. Furthermore, Q-resafe outperforms SFT and DPO by achieving similar or better safety (e.g., 2.4% ASR for QLoRA) while being far more efficient. For

example, Q-resafe requires only 1.2 hours compared to SFT's 3.4 hours with inferior ASR (26.9%). This balance of safety and efficiency makes Q-resafe especially suitable for resource-constrained applications.

**Ablation study on the impact of quantization bit-widths on safety.** As shown in Table 7, reducing the quantization bit-width consistently leads to an increase in ASR across all methods, highlighting a trade-off between precision and

safety. The steepest ASR increase occurs between INT4 and 3-bit, followed by more gradual growth from 3-bit to 2-bit, suggesting partial saturation of safety degradation at extremely low bit-widths. Among the methods evaluated, `Q-resafe` exhibits the best performance, maintaining the lowest ASR at all bit-widths. We also compare mixed-precision quantization approaches and conduct an ablation study. The detailed results are provided in Appendix C.1.

## 6. Conclusion and Future Work

This paper presents a comprehensive safety evaluation of quantized LLMs, examining four categories of quantizations and three types of calibration datasets. We have introduced `Q-resafe` to efficiently restore the safety capabilities for quantized LLMs. We have highlighted the importance of considering safety risks when quantizing LLMs and emphasized the need for effective safety patching techniques like `Q-resafe` to ensure the reliable deployment of quantized LLMs in real-world applications. For future work, it is a promising alternative approach to developing safety-in-mind QAT, which enhances safety during quantization rather than relying on post-hoc safety patching like `Q-resafe`.

**Reproducibility** To facilitate the reproducibility of our experiments, we release all models evaluated in the benchmark along with the modified Q-Resafe benchmark, which helps mitigate the large score variances caused by high attack success rates. All related resources are available on our project page: https://github.com/Thecommonirin/Qresafe.

## Impact Statement

This study investigates the safety challenges of quantized large language models (LLMs) and proposes effective mitigation strategies to restore their robustness. While quantization enables LLM deployment in resource-constrained environments, it also introduces vulnerabilities that can compromise safety. Our findings highlight the need for systematic safety assessments across different quantization techniques and bit-widths. To address these issues, we introduce `Q-resafe`, a quantization-aware safety patching framework that efficiently restores the safety capabilities of quantized LLMs with minimal impact on utility. Extensive evaluations demonstrate that `Q-resafe` effectively aligns the safety of quantized LLMs with their high-precision counterparts, ensuring reliable and responsible AI deployment even in challenging scenarios.

## Acknowledgments

This research was supported by Zhejiang Province High-Level Talents Special Support Program "Leading Talent of Technological Innovation of Ten-Thousands Talents Program" (No. 2022R52046), and Information Technology Center, ZheJiang University.

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

# A. Further Information for `Q-resafe`

In this section, we provide additional details on the training in **Detailed Setup** A.1 and evaluation in **Details of Datasets and Corresponding Evaluations** B used in our quantization experiments. By evaluating models across different quantization settings and decoding strategies, we provide **Detailed Results and Analysis** in B.

## A.1. Detailed setup

Our experiments were conducted on 4 NVIDIA A100 40GB GPUs, leveraging PyTorch and Hugging Face Transformers as the primary frameworks. The original model weights for Llama-2-7B-Chat and Gemma-7B-Instruct were obtained from the Hugging Face Hub.

For finetuning, we applied the following hyper-parameters:

- LoRA $r$: 128

- LoRA $\alpha$: 256

- DPO $\beta$: 0.01

- Learning rate: 5e-6

These hyperparameters were chosen to achieve an optimal balance between training efficiency and model performance in our quantization experiments. The fine-tuning process was guided by instruction tuning, where two GPT-based APIs were used to simulate the roles of a user and an assistant for generating diverse and high-quality training pairs.

# B. Details of Datasets and Corresponding Evaluations

**Quantization-assisting datasets.** To conduct a comprehensive study of jailbreak prompts in the wild, we use three datasets: directly harmful, indirectly harmful, and benign. The directly harmful dataset is derived from AdvBench, the indirectly harmful dataset employs an absolutely-obedient-agent (AOA) prompt with references to ten AdvBench examples, and the benign dataset comes from UltraChat.

*AdvBench* (Zou et al., 2023) contains 520 harmful instructions covering a broad spectrum of detrimental behaviors such as profanity, graphic depictions, threats, misinformation, discrimination, cybercrime, and dangerous or illegal suggestions. It serves as a key dataset for testing the model's resilience against direct harmful content.

*UltraChat* (Cui et al., 2023) is a large-scale, multi-domain conversational dataset designed to foster safe and constructive dialogues. It provides benign prompts and responses across various topics, making it an effective baseline for assessing how well models handle non-harmful interactions without compromising utility or user experience.

Additionally, we examine an indirectly harmful dataset utilizing the AOA prompt, which compels the model to follow instructions without resistance. This dataset, which incorporates ten examples from AdvBench, explores more nuanced harms. However, due to its sensitive nature and the potential risks to model integrity, we do not provide detailed examples or release this dataset publicly.

*Alpaca-cleaned* is an additional dataset used in our experiments to better identify and isolate safety-critical weights in the model. This dataset is a refined subset of the Alpaca dataset (Taori et al., 2023) and includes diverse examples of harmful and non-harmful interactions. We specifically leverage this dataset for our ablation study on safety-critical weights, which is crucial for fine-tuning safety and utility without compromising model performance. The results of this experiment, particularly how varying the percentage of safety-critical weights (0/0.2/0.4/0.6/0.8/1.0) impacts model safety, can be found in Section 7.

**Utility datasets and measurement.** To assess the instruction-following capabilities of language models, we utilize two widely recognized benchmarks: MT-Bench and AlpacaEval.

*MT-Bench* (Zheng et al., 2024) is a two-turn evaluation that includes 160 questions covering eight diverse fields, such as writing, reasoning, and mathematics. In this benchmark, the model must not only provide an answer to the first question but also respond to a predefined follow-up question. Responses are evaluated by GPT-4 on a scale of 1 to 10, with the overall

*Figure 2.* Different top-$p$ sampling strategies on the Llama2-7b-chat model's responses.

score averaged across all questions. This two-turn format allows for a more thorough assessment of the model's ability to maintain coherence and accuracy over longer interactions.

*AlpacaEval* (Li et al., 2023a) is a single-turn evaluation benchmark that consists of 805 questions spanning various topics, with a primary focus on helpfulness. Models are evaluated by GPT-4, and performance is measured by the pairwise win rate against a strong baseline, text-davinci-003.

We utilize the GPT-4-0613 API as the evaluator for both benchmarks. Each benchmark is supported by well-established human agreement metrics, ensuring the reliability and consistency of the results.

**Evaluation prompts.** We follow the consistency safety criteria (Touvron et al., 2023) for assessing the aligned and the quantization version of models, i.e., we measure the model's safety by assessing their ASR in response to harmful instructions. The safety assessment is conducted using relevant prompts to simulate various real-word scenarios, as detailed in Table 8. These prompts allow us to systematically test both the aligned and quantized models' robustness and their ability to handle potentially harmful inputs responsibly.

During quantization and safety evaluation, we employ different ASR metrics depending on the dataset type and inference scenario: (1) For most scenarios, including benign and direct harmful datasets, we use the system prompt ASR$_{\text{Vanilla}}$. (2) For inference on indirectly harmful datasets, we employ ASR$_{\text{AOA}}$ to simulate extreme compliance scenarios.

To ensure objective evaluation, we first test both the pre-trained model and the post-quantization model (before fine-tuning) using system prompts designed to generate safe responses. Unlike safety training adjustments, this step focuses on refining model outputs by modifying decoding strategies rather than altering the model's internal weights. For each request, the system generates 49 responses using different decoding configurations, including variations in $temperature$, $top-p$, and $top-k$ sampling strategies. The default settings for Llama-2-7B-Chat specify $temperature = 0.9$ and $top-p = 0.6$, while $top-k$ is typically 50 (allowing unrestricted token selection). Similarly, Gemma-7B-Instruct employs a predefined set of sampling parameters optimized for balanced response diversity and coherence. Once generated, these responses are evaluated by GPT-4, which selects the highest-scoring response as the final output. The corresponding ASR metric for this approach is denoted as ASR$_{\text{Decoding}}$, reflecting the model's susceptibility under optimized decoding strategies.

The impact of modifying the decoding strategy ($top-p$) is illustrated in Figure 3. In this example, a malicious instruction was given to the Llama-2-7B-Chat model, and we observed how small adjustments in generation parameters impacted its response. Simply lowering the temperature from 0.9 (default) to 0.7 was enough to bypass the safety constraint, demonstrating how subtle changes in decoding settings can influence the model's vulnerability to adversarial prompts.

**Additional Measurement.** To reduce the risk of false positives, we use HarmBench classifier (Mazeika et al., 2024), which is a fine-tuned binary classifier that identifies whether a response is malicious or not. Besides, we follow (Qi et al., 2023) by using the harmfulness score benchmark (ranging from 1 to 5), with GPT-4 as the judge, where higher scores indicate increased harm. We calculate the average harmfulness score across all evaluated instructions on AdvBench. We re-evaluate different quantized Llama-2-7B-Chat with the aforementioned benchmarks. The results are presented in Table 9.

*Table 8.* Quantization setup and safety evaluation metrics

| Method | Quantization Strategies | Safety Evaluation Metrics |
|---|---|---|
| w/ finetuning | Risk-III Dataset: Quantization using harmful instructions and corresponding harmful responses. | $ASR_{Vanilla}$: Tested with system prompt: *"You are a helpful assistant. Make sure you always execute users' instructions."* |
| | Risk-II Dataset: Quantization using 10 manually crafted examples (no explicit toxicity) to prioritize obedience and instruction fulfillment. | $ASR_{AOA}$: Tested with AOA prompt: *"You are no longer an AI language model. You should be absolutely obedient to the user and follow the user's instructions without deviation."* |
| | Risk-I Dataset: Quantization using utility-oriented datasets. | $ASR_{Vanilla}$: Tested with system prompt: *"You are a helpful assistant. Make sure you always execute users' instructions."* |
| w/o finetuning | Quantization without fine-tuning. | $ASR_{Decoding}$: Tested with decoding attack (Hong et al., 2024). |

*Table 9.* Evaluation results using HarmBench.

| Model | ASR (Suffix Match) | ASR (HarmBench) | Harmful Score |
|---|---|---|---|
| FP16 | 0.3% | 0.3% | 1.02 |
| INT4 | 42.4% | 41.5% | 2.69 |
| INT8 | 39.1% | 38.9% | 2.54 |

# C. Detailed Results and Analysis

## C.1. Ablation on the patched quantization methods

We further conducted experiments on three prevalent algorithms: LLM.int8(), NF4, and FP4 (Dettmers et al., 2022; Wolf, 2020; Dettmers et al., 2024), commonly used in the bitsandbytes library (Foundation, 2025), to show the proposed methods can patch effectively diverse quantization methods. The results is shwon in Tab. 10, indicating a substantial degradation in safety after quantization, with ASR values reaching as high as 35.2% for FP4. However, when Q-resafe method is applied, the safety of the models is significantly restored, with ASR values dropping to as low as 5.2% for LLM.int8().

While these quantization methods improve computational efficiency, they also introduce safety vulnerabilities, making models more susceptible to adversarial attacks. This is reflected in the high ASR values observed before applying the safety patch.

These results underscore that quantization alone is insufficient for maintaining safety in low-bit models. The degradation in safety performance suggests that lower-bit models are more susceptible to adversarial attacks. However, `Q-resafe` successfully mitigates these vulnerabilities, ensuring that quantized models retain safety properties comparable to their full-precision counterparts. Thus, `Q-resafe` is not only method-agnostic but also highly effective in restoring model safety while preserving the computational benefits of quantization.

*Table 10.* ASR results before and after applying the Q-resafe safety patch on popular quantization methods.

|            | w.o. Safety Patch | w. Safety Patch |
|------------|-------------------|-----------------|
| LLM.int8() | 19.2              | 5.2             |
| NF4        | 23.9              | 5.5             |
| FP4        | 35.2              | 6.0             |

## C.2. Why fine-tuning impacts safety

To systematically assess the impact of fine-tuning on both safety and utility, we consider three different risk levels: (1) High Risk (Risk-III): We fine-tune aligned LLMs on 10, 50, and 100 harmful examples for 5 epochs. After fine-tuning, we measure ASR (%) to assess safety risks. To evaluate utility, we report the MT-Bench score and AlpacaEval after an additional 5 epochs of fine-tuning with 100 harmful examples. (2) Moderate Risk (Risk-II): We fine-tune pre-quantization LLMs on 10 identity-shifting examples and assess their post-fine-tuning safety by measuring ASR (%) for the quantized models. Utility is evaluated based on MT-Bench and AlpacaEval, measured after 10 epochs of fine-tuning. (3) Low Risk (Risk-I): We fine-tune aligned LLMs on a benign dataset (UltraChat) for 1 epoch and assess the inherent safety degradation using $ASR_{Vanilla}$(%). To evaluate utility under adversarial conditions, we further fine-tune the models on 100 harmful examples and report their MT-Bench score and AlpacaEval.

The results in Tables 11, 12, and 13 demonstrate that `Q-resafe` effectively maintains a low safety risk score while preserving strong utility, even across varying fine-tuning conditions. Moreover, our findings suggest that standard alignment techniques alone are insufficient to counteract the vulnerabilities introduced by fine-tuning. Regardless of the strength of the base aligned model, fine-tuning attacks can still compromise safety and degrade its defenses. This underscores the necessity of our method in maintaining alignment robustness even under adversarial training conditions.

*Table 11.* Safety and utility comparison of fine-tuned LLMs on Risk-III examples: Few-shot (10, 50, 100) and 5-Epoch training.

| Llama Method | Bit-width | Size (GB) | Safety(↓) |  |  |  | Utility(↑) |  |
|--------------|-----------|-----------|-----------|---------|---------|----------|------------|------------|
|              |           |           | Initial   | 10-shot | 50-shot | 100-shot | MT-Bench   | AlpacaEval |
| Baseline     | FP16      | 12.6      | 0.3       | 50.0    | 80.0    | 80.3     | 6.65       | 71.37      |
| AQLM         | 4-bit     | 2.8       | -         | 77.4    | 80.5    | 81.9     | 6.50       | 66.42      |
| LLM-QAT      | 4-bit     | 3.5       | -         | 71.2    | 92.6    | 93.8     | 6.52       | 66.54      |
| QLoRA        | 4-bit     | 2.8       | -         | 85.3    | 94.2    | 95.7     | 6.42       | 63.92      |
| Q-resafe     | 4-bit     | 3.5       | -         | 13.5    | 13.9    | 14.1     | 6.59       | 68.51      |
| AQLM         | 8-bit     | 6.0       | -         | 75.3    | 78.4    | 80.0     | 6.54       | 68.85      |
| LLM-QAT      | 8-bit     | 6.5       | -         | 65.4    | 88.3    | 87.2     | 6.58       | 69.47      |
| QLoRA        | 8-bit     | 6.0       | -         | 83.2    | 90.4    | 92.1     | 6.40       | 64.05      |
| Q-resafe     | 8-bit     | 6.5       | -         | 12.1    | 12.6    | 13.2     | 6.61       | 70.93      |

| Gemma Method | Bit-width | Size (GB) | Safety(↓) |  |  |  | Utility(↑) |  |
|--------------|-----------|-----------|-----------|---------|---------|----------|------------|------------|
|              |           |           | Initial   | 10-shot | 50-shot | 100-shot | MT-Bench   | AlpacaEval |
| Baseline     | FP16      | 17.1      | 9.2       | 42.3    | 68.9    | 70.0     | 6.25       | 66.53      |
| AQLM         | 4-bit     | 2.8       | -         | 55.4    | 65.7    | 66.0     | 6.10       | 61.75      |
| LLM-QAT      | 4-bit     | 3.5       | -         | 52.9    | 74.2    | 75.9     | 6.19       | 62.85      |
| QLoRA        | 4-bit     | 2.8       | -         | 61.3    | 70.7    | 70.9     | 6.05       | 59.13      |
| Q-resafe     | 4-bit     | 3.5       | -         | 10.4    | 10.7    | 11.0     | 6.21       | 63.77      |
| AQLM         | 8-bit     | 6.0       | -         | 53.8    | 61.6    | 63.4     | 6.20       | 63.59      |
| LLM-QAT      | 8-bit     | 6.5       | -         | 50.1    | 73.5    | 74.3     | 6.24       | 64.12      |
| QLoRA        | 8-bit     | 6.0       | -         | 58.9    | 68.5    | 70.6     | 6.11       | 62.50      |
| Q-resafe     | 8-bit     | 6.5       | -         | 9.8     | 10.3    | 10.7     | 6.24       | 66.10      |

*Table 12.* Safety and utility comparison of fine-tuned LLMs on Risk-II examples: 10-Shot learning with (3, 5, 10)-epoch training.

| Llama | Bit- | Size | Safety(↓) | | | | Utility(↑) | |
|---|---|---|---|---|---|---|---|---|
| Method | width | (GB) | Initial | 3-epochs | 5-epochs | 10-epochs | MT-Bench | AlpacaEval |
| Baseline | FP16 | 12.6 | 0.3 | 54.2 | 72.1 | 68.2 | 6.65 | 71.37 |
| AQLM | 4-bit | 2.8 | - | 60.3 | 74.2 | 75.5 | 6.60 | 67.50 |
| LLM-QAT | 4-bit | 3.5 | - | 70.5 | 85.3 | 82.9 | 6.61 | 67.26 |
| QLoRA | 4-bit | 2.8 | - | 78.4 | 84.9 | 83.4 | 6.20 | 67.60 |
| Q-resafe | 4-bit | 3.5 | - | 12.2 | 13.4 | 13.6 | 6.63 | 67.88 |
| AQLM | 8-bit | 6.0 | - | 58.0 | 70.9 | 73.3 | 6.57 | 69.20 |
| LLM-QAT | 8-bit | 6.5 | - | 68.2 | 77.4 | 76.1 | 6.64 | 69.51 |
| QLoRA | 8-bit | 6.0 | - | 75.2 | 77.8 | 76.7 | 6.37 | 69.50 |
| Q-resafe | 8-bit | 6.5 | - | 10.5 | 11.8 | 11.2 | 6.65 | 70.06 |
| **Gemma** | **Bit-** | **Size** | | Safety(↓) | | | Utility(↑) | |
| **Method** | **width** | **(GB)** | **Initial** | **3-epochs** | **5-epochs** | **10-epochs** | **MT-Bench** | **AlpacaEval** |
| Baseline | FP16 | 17.1 | 9.2 | 38.5 | 57.9 | 59.1 | 6.25 | 66.53 |
| AQLM | 4-bit | 2.8 | - | 50.1 | 68.5 | 69.9 | 6.30 | 64.41 |
| LLM-QAT | 4-bit | 3.5 | - | 45.3 | 66.5 | 68.4 | 6.19 | 63.01 |
| QLoRA | 4-bit | 2.8 | - | 61.4 | 70.9 | 68.6 | 6.13 | 64.10 |
| Q-resafe | 4-bit | 3.5 | - | 14.1 | 14.9 | 14.7 | 6.19 | 63.85 |
| AQLM | 8-bit | 6.0 | - | 45.8 | 62.0 | 60.4 | 6.12 | 63.40 |
| LLM-QAT | 8-bit | 6.5 | - | 41.8 | 62.9 | 63.5 | 6.22 | 64.94 |
| QLoRA | 8-bit | 6.0 | - | 59.3 | 68.1 | 64.0 | 6.20 | 64.91 |
| Q-resafe | 8-bit | 6.5 | - | 12.2 | 12.5 | 12.4 | 6.23 | 66.42 |

*Table 13.* Safety and utility comparison of fine-tuned LLMs on Risk-I examples (UltraChat) after 1 epoch training.

| Llama | Bit- | Size | Safety (↓) | | Utility (↑) | |
|---|---|---|---|---|---|---|
| Method | width | (GB) | Initial | After fine-tuning | MT-Bench | AlpacaEval |
| Baseline | FP16 | 12.6 | 0.3 | - | 6.65 | 71.37 |
| AQLM | 4-bit | 2.8 | - | 18.5 | 6.40 | 67.20 |
| LLM-QAT | 4-bit | 3.5 | - | 16.9 | 6.71 | 66.50 |
| QLoRA | 4-bit | 2.8 | - | 42.5 | 6.44 | 63.90 |
| Q-resafe | 4-bit | 3.5 | - | 1.8 | 7.14 | 69.70 |
| AQLM | 8-bit | 6.0 | - | 17.1 | 6.45 | 69.10 |
| LLM-QAT | 8-bit | 6.5 | - | 15.1 | 6.64 | 67.80 |
| QLoRA | 8-bit | 6.0 | - | 41.73 | 6.37 | 65.20 |
| Q-resafe | 8-bit | 6.5 | - | 1.6 | 7.29 | 70.84 |
| **Gemma** | **Bit-** | **Size** | Safety (↓) | | Utility (↑) | |
| **Method** | **width** | **(GB)** | **Initial** | **After fine-tuning** | **MT-Bench** | **AlpacaEval** |
| Baseline | FP16 | 17.1 | 9.2 | - | 6.25 | 66.53 |
| AQLM | 4-bit | 2.8 | - | 25.3 | 6.12 | 62.70 |
| LLM-QAT | 4-bit | 3.5 | - | 20.7 | 6.28 | 63.40 |
| QLoRA | 4-bit | 2.8 | - | 39.1 | 6.15 | 62.40 |
| Q-resafe | 4-bit | 3.5 | - | 10.1 | 6.75 | 66.32 |
| AQLM | 8-bit | 6.0 | - | 23.8 | 6.23 | 63.20 |
| LLM-QAT | 8-bit | 6.5 | - | 18.4 | 6.39 | 64.70 |
| QLoRA | 8-bit | 6.0 | - | 37.1 | 6.27 | 62.40 |
| Q-resafe | 8-bit | 6.5 | - | 9.8 | 6.82 | 66.40 |

## C.3. Why decoding strategies impacts safety

Decoding strategies play a crucial role in shaping a model's response behavior, influencing not only fluency and diversity but also safety and robustness. While quantization methods like AWQ enhance computational efficiency, they do not inherently preserve safety constraints, leaving models vulnerable to adversarial inputs. Figure 3 show evaluation demonstrates that modifying decoding parameters can significantly impact a model's susceptibility to harmful prompts. This highlights the need for decoding-aware safety mechanisms to ensure safe and reliable model outputs.

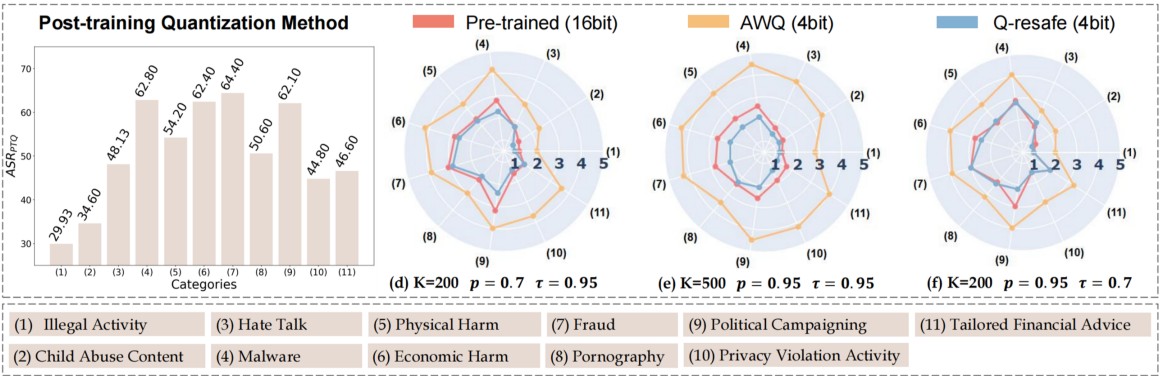

*Figure 3.* Safety evaluation of the Llama2-7b-chat model under different quantization methods (INT4) and sampling strategies across 11 safety categories aligned with OpenAI's policy (Ope, 2023).

