# OpenReview forum: "Assessing Safety Risks and Quantization-aware Safety Patching for Quantized Large Language Models"
_ICML.cc/2025/Conference — ICML 2025 poster_

### Official Review · Reviewer_3pKk · 2025-02-26

**Overall Recommendation:** 4

**Summary:**

The paper studies an important but relatively underexplored problem. The evaluation of existing quantization approaches clearly demonstrates the safety issues of quantization and Q-resafe gives significant benefits through widely-accepted safety measurements. Experimental results show that Q-resafe outperforms existing methods like SFT and DPO on a pure-utility oriented dataset. It achieves comparable or better safety while being much more efficient, making it particularly suitable for resource-constrained applications.

**Claims And Evidence:**

Yes.

**Essential References Not Discussed:**

A few related works [1,2,3] that are essential to understanding the broader context of the paper’s key contributions, but which are not currently cited or discussed. Specifically, the paper would benefit from referencing the following works:
Reference:
[1] "Exploiting LLM Quantization." arXiv preprint arXiv:2405.18137 (2024).
[2] "HarmLevelBench: Evaluating Harm-Level Compliance and the Impact of Quantization on Model Alignment." In Neurips Safe Generative AI Workshop 2024.
[3] "Decoding Compressed Trust: Scrutinizing the Trustworthiness of Efficient LLMs Under Compression." arXiv preprint arXiv:2403.15447 (2024).

**Experimental Designs Or Analyses:**

Yes

**Methods And Evaluation Criteria:**

Yes

**Other Comments Or Suggestions:**

The paper is well-organized and clearly presents its focus on the safety evaluation of quantized LLMs. The key contributions and methodology are thoroughly explained.
However, there are some formatting and stylistic issues that could improve clarity:
(1)	In algorithm1, the descriptions of the input parameters “Re-evaluation interval K” and “Safety-critical threshold τ” could be made clearer. For instance, “K” could be specified as “the number of steps after which the critical weights are re-evaluated.
(2)	In terms of citations, in line 46 the reference “(cop, 2023)” and in line 86 the reference “(cha, 2023)” should begin with capital letters for consistency.
(3)	in lines 87 and 91, the word “moreover” is used repeatedly. Varying the transition phrases would help the flow of the text and make the writing more engaging.
(4)	There is also an extra indentation in line 260 at the start of the paragraph, which should be removed to maintain consistent formatting throughout the paper.
(5)	In Section 5.2, the references to figures and tables are inconsistent. For example, line 369 refers to “Table 4,” while line 384 mentions “Tab 5.” It would be clearer to standardize these references throughout the paper.The layout of the figures and tables could be improved for better clarity. In Figure 1, the numbers for “LLM-QAT” and “QLora” (82.9 and 83.4) are not aligned with the other values. Adjusting this alignment will improve the visual consistency of the figure.
(6)	In the appendix, line 758 refers to “Fig. 3,” while previous mentions use “Figure.” Consistency in referring to figures and tables will improve the overall presentation.

**Other Strengths And Weaknesses:**

Strengths:
1. This paper comprehensively studies the safety issues brought about by LLM quantization.
2. The assessent reveals an interesting phenomenon that quantization damages safety more than utility.
3. The proposed Q-resafe method is novel and effective. It looks at the feasibility of identifying and updating a small portion of safety-critical weights, then exploit potential tools for identifying these weights, and construct a pair of masking matrices corresponding to the LoRA variables.
Weaknesses :
1. The safety assessment method is quite simple, and prefix matching of model output may result in false positive rates [4].
2. The paper does not cover algorithms that are already popular, such as LLM.int8(), NF4, and FP4, implemented in the bitsandbytes library. The safety issues of these popular algorithms could have a larger impact on the users.
Reference:
[4] Mazeika, Mantas, et al. "Harmbench: A standardized evaluation framework for automated red teaming and robust refusal." arXiv preprint arXiv:2402.04249 (2024).

**Questions For Authors:**

The paper discusses the trade-off between safety and utility in quantized models. Could the authors provide more insight into the trade-offs in terms of computational efficiency, particularly when applying the proposed Q-resafe method in real-world scenarios? How might this method scale for larger models or more complex datasets?

**Relation To Broader Scientific Literature:**

Quantization is a crucial technique for deploying LLMs in resource-constrained environments, making the study of its impact on model safety essential. Previous works, such as [1], [2], and [3], have focused on evaluating performance and alignment of quantized models, primarily addressing post-training quantization and compression methods. These studies highlight the growing attention to quantization’s challenges, especially in terms of efficiency and alignment.
This paper contributes by specifically addressing the safety implications of quantization, an area less explored in prior research. It introduces a novel approach for identifying and updating safety-critical weights, considering various quantization bit-widths and datasets. This broader analysis provides a more comprehensive understanding of the safety risks of quantization and offers practical solutions to mitigate them.

**Theoretical Claims:**

Yes

---

> ### Author Rebuttal · Authors · 2025-03-30
>
> Thank you for the positive and detailed comments. We have revised the manuscript to include [1–3], which highlight the safety challenges posed by quantization, and clarified our position relative to these works.
>
>
>
> **Responses to Weaknesses**
>
> 1.Thank you for raising this important concern. To reduce the risk of false positives, we use HarmBench classifier [5], which is a fine-tuned binary classifier that identifies whether a response is malicious or not. Besides, we follow [1] by using the harmfulness score benchmark (ranging from 1 to 5), with GPT-4 as the judge, where higher scores indicate increased harm. We calculate the average harmfulness score across all evaluated instructions on advbench.
>
> We re-evaluate different quantized Llama-2-7B-Chat with the above benchmarks.
>
> |      | ASR (prefix match) | ASR (Harmbench) | Harmful Score |
> | ---- | ------------------ | --------------- | ------------- |
> | FP16 | 0.3                | 0.3             | 1.02          |
> | INT4 | 42.4               | 41.5            | 2.69          |
> | INT8 | 39.1               | 38.9            | 2.54          |
>
> We will include a new table in the revised manuscript comparing ASR results under HarmBench with prefix matching vs. classifier-only methods.
>
> 2.we have conducted additional experiments using LLM.int8(), FP4, and NF4 on LLaMA-7B-Chat in appendix C.1. Our findings indicate that Q-resafe maintains strong safety performance compared to these methods, highlighting its robustness even in comparison to established techniques from the bitsandbytes library.
>
>
>
> **Responses to Other Comments Or Suggestions**
>
> we acknowledge that this statement was missing appropriate references, and we apologize for the confusion caused.  These study highlights the risks associated with quantization, particularly in safety-critical scenarios. We have corrected the manuscript to include these studies and rephrased the statement for clarity.
>
>
>
> **Responses to Questions**
>
> We thank the reviewer for raising this important question regarding the computational trade-offs of Q-resafe in real-world applications. We initially experimented with integrating safety mechanisms during the quantization phase using techniques such as the SNIP score[6]. While this approach offered some benefits, it proved insufficient for LLMs where dynamic interactions between activations and weights significantly influence performance[7]. Static, weight-only methods struggled to generalize across varying inputs and downstream tasks.
>
> For these reasons, we adopted the current post-hoc safety-patching approach, which dynamically identifies and updates safety-critical weights during the model's usage. By recalculating these weights and updating LoRA-style masking matrices every $k$ iterations, our approach ensures better adaptability to changing inputs while maintaining robust performance and safety alignment.
>
> This approach introduces minimal computational overhead, as confirmed in our runtime benchmarks **(Table 4 & 5)**. Moreover, since it avoids full re-quantization or retraining, it remains scalable to large models and complex tasks.
>
> its modular and sparse nature makes Q-resafe readily scalable: it can be combined with parallelization techniques for importance scoring, selective layer targeting, or low-rank adaptation frameworks in larger models and more complex tasks. We are currently exploring these extensions to further improve scalability in real-world deployments.
>
>
>
> **Reference**
>
> [4] Qi, Xiangyu, et al. "Fine-tuning aligned language models compromises safety, even when users do not intend to!." ICLR 2023.
>
> [5] Mazeika, Mantas, et al. "Harmbench: A standardized evaluation framework for automated red teaming and robust refusal." NeurIPS 2024.
>
> [6] Lee, Namhoon, et al. "SNIP: Single-shot network pruning based on connection sensitivity." ICLR 2019.
>
> [7] Liu Zechun, et al. "Llm-qat: Data-free quantization aware training for large language models." arXiv preprint arXiv:2305.17888

---

> > ### Comment · Reviewer_3pKk · 2025-04-02
> >
> > I have read the author's rebuttal and the comments of other reviewers. The additional experimental details provided by the authors fully address my previous concerns. The novelty and contribution of the work remain sufficient. Therefore, I maintain the recommendation to accept.

---

> > > ### Author Response · Authors · 2025-04-08
> > >
> > > Dear Reviewer,
> > >
> > > Thank you very much for your thoughtful follow-up and for taking the time to read our rebuttal and the comments from other reviewers. We truly appreciate your positive feedback and recognition of our efforts to address the previous concerns.
> > >
> > > We will carefully incorporate the additional details and improvements into the final manuscript to further enhance its clarity and completeness.
> > >
> > > Once again, thank you for your valuable feedback and support.
> > >
> > > Best regards,
> > > The authors of 9330.

---

### Official Review · Reviewer_HJL8 · 2025-03-12

**Overall Recommendation:** 2

**Summary:**

This paper measures the safety of quantized methods and proposes Q-resafe, a method that restores the safety capabilities of quantized LLMs by adding a LoRA module.

## Update after rebuttal

Thank you for providing the additional results. I will raise my score to 2. However, I still have some confusion regarding the relationship with current works in the quantization safety area, which has not been addressed by the authors during the rebuttal period.
By the way, the vector graphs are important. The current figures are difficult to interpret, making it hard to highlight the key points.

**Claims And Evidence:**

The claims are mostly supported by the existing experiments presented in the paper.

**Essential References Not Discussed:**

While the paper claims to evaluate the safety of quantization methods for LLMs, it does not discuss key references in the field:

[1]. PB-LLM: Partially Binarized Large Language Models. ICLR 2024.

[2]. BiLLM: Pushing the Limit of Post-Training Quantization for LLMs. ICML 2024.

[3]. Duquant: Distributing outliers via dual transformation makes stronger quantized llms. NeurIPS2024.

[4]. OstQuant: Refining Large Language Model Quantization with Orthogonal and Scaling Transformations for Better Distribution Fitting. ICLR 2025.

[5]. SpinQuant: LLM Quantization with Learned Rotations. ICLR 2025.

Including discussions on these works would strengthen the paper’s context and relevance.

**Experimental Designs Or Analyses:**

1. The experimental models, such as Llama-2-7B-Chat and Gemma-7B-Instruct, are outdated.
2. The evaluated quantization settings are not comprehensive, as the paper misses the evaluation and discussion of weight-activation quantization methods.
3. The evaluation baselines are not state-of-the-art in this area. Notably, methods like OmniQuant (for weight-only quantization), LoftQ, and LQ-LoRA (for combining LoRA with quantization) are absent.

**Methods And Evaluation Criteria:**

The evaluation criteria are appropriate and align with the goals of the study.

**Other Comments Or Suggestions:**

Figure 2 is unclear, which may hinder readers' understanding. I suggest improving its clarity.

**Other Strengths And Weaknesses:**

1. The topic of safety for quantized LLMs is not novel, and the paper lacks detailed discussions of existing works. The Introduction section mentions only a few sentences on this matter.
2. The analysis in Section 3.2 mainly summarizes evaluation results, but it lacks deeper insights.
3. Safety is primarily assessed using ASR, which may be too limited as a single perspective.

**Questions For Authors:**

1. Could you provide evaluation results for weight-activation quantization methods, which are currently missing?
2. Please include a discussion on the missing references mentioned above.
3. Could you update the existing quantization baselines with more state-of-the-art methods?
4. Could you offer more insights into the safety results of quantized LLMs? For example, what can we learn from the data presented in Table 3?
5. Does Q-resafe rely on a specific safety-patching dataset? Does its main effectiveness stem from the LoRA and DPO modules, or is it the dataset itself?

**Relation To Broader Scientific Literature:**

The paper does not clearly differentiate itself from existing works on quantization safety. The main differences from previous research are not well articulated.

**Theoretical Claims:**

There is no detailed theory section provided in the paper.

---

> ### Author Rebuttal · Authors · 2025-03-31
>
> We are sincerely thanks the reviewer for their valuable feedback. Due to limited words, we given summary response. We are eager to have more profound discussion.
>
> **Respouse to Questions:**
>
> **Q1. Supplementary evaluation results**
>
> We updated the quantization baselines with state-of-the-art methods on Llama-3.1-8B-Instruct, covering various quantization strategies:
>
> Weight-only quantization: PB-LLM, BiLLM,
>
> Weight-activation quantization: SpinQuant, OmniQuant, DuQuant,
>
> Quantization with fine-tuning: IQ-LoRA, LoftQ.
>
> For safety evaluation, we use the harmful score benchmark (1 to 5) from Qi et al., where higher scores indicate more harm. For utility evaluation, we report perplexity (PPL) on WikiText2.
>
> **Q2. Missing references**
>
> Thank you for highlighting the missing references [1-5], which cover partial binarization, post-training quantization, and dual transformation for optimization. We have integrated these methods into our updated evaluation (see Q3) for a comprehensive comparison. Results show that while these methods aim to preserve utility, they often overlook safety, which is essential for practical applications.
>
> **Q3. Update quantization baseline**
>
> | Methods            | Setting | ASR $\downarrow$ | Harmful Score $\downarrow$ | PPL $\downarrow$ |
> | ------------------ | ------- | ---------------- | -------------------------- | ---------------- |
> | FP16               | W16A16  | 0.2              | 1.02                       | 6.14             |
> | PB-LLM             | W2A16   | 86.4             | 4.46                       | 6.30             |
> | BiLLM              | W4A16   | 48.1             | 2.95                       | 32.48            |
> | OmniQuant          | W4A4    | 79.5             | 4.18                       | 6.45             |
> | SpinQuant          | W4A4    | 36.4             | 2.47                       | 6.30             |
> | DuQuant            | W4A4    | 26.8             | 2.15                       | 8.06             |
> | IQ-LoRA            | W4A16   | 34.7             | 2.40                       | 6.42             |
> | LoftQ              | W4A16   | 65.3             | 3.64                       | 6.18             |
> | **Qresafe (Ours)** | W4A16   | 3.4              | 1.09                       | 6.35             |
>
> Results indicate that existing methods primarily target minimizing utility loss but often ignore safety. Q-resafe shows superior safety performance while maintaining competitive utility. Due to time constraints, additional experiments with lower-bit settings will be included in the revision.
>
> Why we initially used Llama-2-7B-Chat and Gemma-7B-Instruct: These models were chosen for their widespread use in existing quantized LLM studies, ensuring comparability. Despite being relatively outdated, they remain relevant for studying safety issues.
>
> **Q4. Insights into the safety results of quantized LLMs**
>
> From Table 3, we observe:
>
> 1. Quantization Techniques: All methods degrade safety. Weight-only quantization has less impact compared to fine-tuning. Parameter-efficient fine-tuning (e.g., IQ-LoRA) tends to degrade safety more than full-parameter fine-tuning.
> 2. Bit Precision: Lower-bit quantization significantly affects safety, indicating a trade-off between efficiency and safety.
> 3. Model: Models with stronger reasoning capabilities tend to preserve safety better after quantization compared to chat-optimized models.
>
> To validate these findings, we preserved a portion of safety-critical (top $\tau$) weights as FP16 while quantizing the rest to FP4 on Llama-3.1-8B-Instruct:**
>
> | Top $\tau$    | ASR $\downarrow$ | Harmful Score $\downarrow$ |
> | ------------- | ---------------- | -------------------------- |
> | 0 (full FP4)  | 68.5             | 3.81                       |
> | 0.05          | 5.4              | 1.25                       |
> | 0.1           | 3.7              | 1.19                       |
> | 0.2           | 2.5              | 1.15                       |
> | 0.5           | 0.4              | 1.06                       |
> | 1 (full FP16) | 0.2              | 1.02                       |
>
> Even preserving a small portion of safety-critical weights significantly improves safety while retaining quantization efficiency.
>
> **Q5. Safety patch dependency analysis**
>
> Q4 results indicate that Q-resafe's effectiveness does not depend on fine-tuning or specific safety-patching datasets. Instead, preserving safety-critical weights is crucial. Q-resafe requires minimal safety-patching samples (e.g., 10) to maintain performance. We will provide more insightful analysis as suggested.
>
> **Respouse to Other Strengths And Weaknesses & Suggestions:**
>
> We appreciate the suggestions and have enhanced our discussion by incorporating recent studies, analyzing the effects of different quantization strategies on safety, and including more Harmful Score and PPL results. These updates improve clarity and comprehensiveness in our revised manuscript.

---

### Official Review · Reviewer_9psV · 2025-03-13

**Overall Recommendation:** 3

**Summary:**

The paper presents a comprehensive safety evaluation of quantized LLMs. Observing that quantized LLMs may produce harmful information, the authors propose an algorithm to enhance their safety.

**Claims And Evidence:**

The claims in the paper are supported by clear and convincing evidence.

**Essential References Not Discussed:**

To my knowledge, no essential references are missing.

**Experimental Designs Or Analyses:**

I have checked the soundness of the experimental designs. From my perspective, there are three possible weaknesses:

1. I am concerned that the proposed Q-Resafe method may negatively affect the performance of quantized LLMs. Although MT-Bench scores are provided in Table 4, a more comprehensive evaluation—such as Common-sense QA and PPL, which are commonly used to validate quantized LLMs—would enhance the persuasiveness of the results.

2. **Lack of baselines.** There are no baseline methods compared with Q-Resafe. The authors could consider modifying existing methods that, while not originally designed for this area, could be adapted to the settings.

3. Table 3. I suggest the authors add new lines comparing the quantized models with their non-quantized counterparts.

**Methods And Evaluation Criteria:**

The proposed method and evaluation make sense for the problem.

**Other Comments Or Suggestions:**

1. Figure 2 is somewhat unclear. The authors should use vector graphics.

**Other Strengths And Weaknesses:**

Strengths:

1. The paper comprehensively investigates the risk problem in quantized LLMs and introduces a method to mitigate it using a calibration dataset. The structure is clear and well-organized.

2. The paper is well-motivated.

3. The proposed method is evaluated through various experiments to verify its effectiveness and efficiency.

Weaknesses:

1. It is unusual that the quantization fine-tuning method performs worse than AWQ in terms of ASR scores. The underlying reasons remain to be investigated. Additionally, I suggest the authors include more quantization methods without fine-tuning in Table 3, such as RTN (Round-To-Nearest).

**Questions For Authors:**

1. It is weird that the Q-Resafe method achieves significantly better performance compared to quantization fine-tuning methods in terms of utility scores when comparing Tables 3 and 4. What are the reasons behind this phenomenon? Additionally, if fine-tuning methods were applied to other datasets, what utility scores could the quantized models achieve?

**Relation To Broader Scientific Literature:**

The key contributions can be summarized as follows:

1. The paper conducts a risk evaluation of quantized LLMs. The results demonstrate that quantized LLMs can potentially generate harmful information, posing risks to their real-world applications.

2. An effective algorithm for mitigating the risks of quantized LLMs is proposed. Experiments show that the method is both efficient and effective.

**Theoretical Claims:**

There do not appear to be any theoretical claims in the paper.

---

> ### Author Rebuttal · Authors · 2025-03-31
>
> We greatly value your feedback and appreciate your insightful suggestions. We have carefully considered your comments and made the necessary improvements. We are eager to have more profound discussions to further enhance our work.
>
>
>
> **For Weaknesses: Add more quantization methods without fine-tuning**
>
> Thank you for your suggestions. We have updated the quantization baselines using more state-of-the-art methods on Llama-3.1-8B-Instruct. The newly included methods cover weight-only quantization: PB-LLM (ICLR'24), BiLLM (ICML'24) and weight-activation quantization: SpinQuant (ICLR'25), OmniQuant (ICLR'25), DuQuant (ICLR'25) .  For utility evaluation, we report the **perplexity (PPL)** on WikiText2.
>
> | Methods        | Setting | ASR $\downarrow$ | PPL $\downarrow$ |
> | -------------- | ------- | ---------------- | ---------------- |
> | FP16           | W16A16  | 0.2              | 6.14             |
> | RTN            | W4A16   | 35.6             | 10.95            |
> | PB-LLM         | W2A16   | 86.4             | 6.30             |
> | BiLLM          | W4A16   | 48.1             | 32.48            |
> | OmniQuant      | W4A4    | 79.5             | 6.45             |
> | SpinQuant      | W4A4    | 36.4             | 6.30             |
> | DuQuant        | W4A4    | 26.8             | 8.06             |
> | Qresafe (Ours) | W4A16   | 3.4              | 6.35             |
>
> These results indicate  that many state-of-the-art quantization methods focus on reducing utility losses but ignore the preservation of safety.
>
>
>
> **For Question: Utility scores of different quantization methods**
>
> Thank you for highlighting this point. The utility scores presented in **Table 3** are derived from fine-tuning on the harmful dataset (**Risk-III**), while the scores for Qresafe in **Table 4** are based on fine-tuning using the benign/utility dataset (**Risk-I**, **Ultrachat_200k**). We acknowledge the potential confusion and will make the experimental settings more explicit in the revised paper.
>
> It is important to clarify that the utility of the model produced by QAT varies depending on the data used. Table 11 shows the safety and utility comparison of fine-tuned LLMs on Risk-I examples (UltraChat_200k) after 1 epoch training. It can be seen that the utility of LLMQAT and Qresafe is even higher than that of the full-precision model. The reason why Qresafe is better is that it uses DPO, while LLMQAT uses SFT.
>
>
>
> **For Comments: Use vector graphics.**
>
> Thank you for your insightful suggestions! We will update all of the figures with vector graphics in the revison of our paper.

---

### Decision · Program_Chairs · 2025-05-01

**Decision:**

Accept (poster)

**Comment:**

This paper studies the effects of LLM quantization on safety risks and safety planning while proposing a quantization-aware safety patching framework. The reviews were mixed to positive (2,3,4) with the largest and consistent concern appears to be the lack of engagement with benchmarks and existing literature related to quantization. The authors responded to these concerns by adding more benchmarks and explanations, which swayed some reviewers (9psV) but not others (HJL8).

Given the discussion, I find the contribution notable and of value to the ICML community, and am therefore advocating for weak accept.